# Benefits and Costs of a Community-Led Total Sanitation Intervention in Rural Ethiopia—A Trial-Based Ex Post Economic Evaluation

**DOI:** 10.3390/ijerph17145068

**Published:** 2020-07-14

**Authors:** Seungman Cha, Sunghoon Jung, Dawit Belew Bizuneh, Tadesse Abera, Young-Ah Doh, Jieun Seong, Ian Ross

**Affiliations:** 1Faculty of Infectious and Tropical Disease, London School of Hygiene & Tropical Medicine, Keppel Street, London WC1E 7HT, UK; Ian.Ross@lshtm.ac.uk; 2Department of Global Development and Entrepreneurship, Graduate School of Global Development and Entrepreneurship, Handong Global University, Pohang 37554, Korea; 3Good Neighbors International, Mozambique, Maputo, Mozambique; hoonie12@gmail.com; 4Independent Consultant, Addis Ababa, Ethiopia; belewbizuneh@gmail.com; 5Public Health Institute, Addis Ababa, Ethiopia; tade842@gmail.com; 6Korea International Cooperation Agency, Seongnam 13449, Korea; yadoh@koica.go.kr (Y.-A.D.); sje0115@koica.go.kr (J.S.)

**Keywords:** cost–benefit analysis, community-led total sanitation, Ethiopia, sanitation improvements, household latrine

## Abstract

We estimated the costs and benefits of a community-led total sanitation (CLTS) intervention using the empirical results from a cluster-randomized controlled trial in rural Ethiopia. We modelled benefits and costs of the intervention over 10 years, as compared to an existing local government program. Health benefits were estimated as the value of averted mortality due to diarrheal disease and the cost of illness arising from averted diarrheal morbidity. We also estimated the value of time savings from avoided open defecation and use of neighbours’ latrines. Intervention delivery costs were estimated top-down based on financial records, while recurrent costs were estimated bottom-up from trial data. We explored methodological and parameter uncertainty using one-way and probabilistic sensitivity analyses. Avoided mortality accounted for 58% of total benefits, followed by time savings from increased access to household latrines. The base case benefit–cost ratio was 3.7 (95% CI: 1.9–5.4) and the net present value was Int’l $1,193,786 (95% CI: 406,017–1,977,960). The sources of the largest uncertainty in one-way sensitivity analyses were the effect of the CLTS intervention and the assumed lifespan of an improved latrine. Our results suggest that CLTS interventions can yield favourable economic returns, particularly if follow-up after the triggering is implemented intensively and uptake of improved latrines is achieved (as opposed to unimproved).

## 1. Introduction

Universal access to safely managed sanitation is one of the sustainable development goal targets [1,2]. However, progress has been slow in sub-Saharan Africa, where access to safely managed sanitation services expanded from 15% to 18% between 2000 and 2017 [2]. By contrast, it increased from 32% to 64% in South Eastern Asia [2].

The community-led total sanitation (CLTS) approach was initiated in Bangladesh in 2000 [3]. It attempts to motivate behaviour change by triggering a collective sense of disgust about open defecation [4]. A core principle of CLTS is not to provide subsidies for latrine construction, but there has been debate about whether to provide subsidies to the poorest and most marginalized [5,6]. CLTS was also criticized as promoting unimproved latrines, and a recent study suggested that unimproved latrines might lead to reversion to open defecation [7,8]. In response to these critics, the CLTS approach has evolved, sometimes now implemented in combination with sanitation marketing or subsidies to the poorest households [6]. More than 30 countries have adopted CLTS as a national sanitation policy, including Ethiopia [6]. After a pilot sanitation program with a community-led approach in 2007, the Ethiopian government developed a set of guidelines and verification protocols, adding a hygiene component in 2008 (called CLTSH) [9].

Economic evaluations inform investment decisions by comparing costs and outcomes of investment options [10]. Understanding the benefits and costs of sanitation interventions can help make informed decisions about resource allocation, in particular in identifying settings where there is high likelihood of a net benefit [11].

According to a recent review of the knowledge base for sanitation interventions, most cost–benefit analyses (CBAs) were identified as presenting global-level and/or ex-ante analyses, relying heavily on assumptions for parameter values [12]. The authors suggested that more ex-post CBA studies using empirical data from interventions settings to inform model parameters are required. Other studies have also emphasized that costs and benefits can be highly context-specific, arguing for more primary studies [13,14,15]. Radin and colleagues argued that “there is little more to be learned from global desk-top benefit–cost calculations of sanitation interventions” because the outcomes are dependent on local context [16]. They reconfirmed the necessity of CBA based on primary data collection. Although the benefit and costs of CLTS or sanitation interventions are context-specific, a cost–benefit analysis could help to select a site for a future project and to evaluate the performance of an intervention if the analysis is done at a specific site [17,18,19]. Empirical data in a specific context can also provide the best-available evidence for country- or global-level cost–benefit analyses [17,18,19].

The few existing ex-post studies do not present tables of parameter values, or equations setting out how they calculated costs and benefits, precluding transparency and replication [20,21,22].

Radin and colleagues argued that the benefits of CLTS interventions in prior studies had been overestimated largely because the time commitment of community members mobilized for the intervention was not taken into account [16]. Their base case benefit–cost ratio (BCR) was 1.7 for medium-uptake villages. They found that, while the net present value was positive in 75% of the simulations under sensitivity analysis, CLTS was unlikely to be that attractive compared to the interventions in other sectors with far higher BCRs [16].

However, their estimation was done on a hypothetical basis, rather than using model parameters informed by primary data collection. Given these recent reviews’ and studies’ emphasis on the need for more ex-post and empirical CBA studies, we designed the present study. We aim to estimate the costs and benefits of a CLTS intervention using the primary data from a cluster-randomized controlled trial of a CLTS intervention in rural areas of Ethiopia. To our knowledge, this is the first ex post CBA of a CLTS intervention based on actual implementation in a given context [23]. For transparency, we present detailed parameter tables and equations demonstrating how costs and benefits were calculated.

## 2. Materials and Methods

### 2.1. Study Area

This cost–benefit analysis (CBA) took place alongside a randomized trial, the protocol and results of which are reported elsewhere [24,25]. The water and sanitation intervention of the Korea International Cooperation Agency (KOICA) took place in the Southern Nations, Nationalities, and Peoples’ Region (SNNPR), a state in South-Western Ethiopia, specifically in the Cheha and the Enemor Ena Ener woredas (districts). The total population of these predominantly rural districts was 133,233 and 204,937, respectively, in 2014 [26]. Overall water and sanitation intervention took place in 212 gotts (villages) distributed across all kebeles (sub-districts) within those districts, of which 48 gotts were included in the trial. All other interventions except for CLTS were implemented after the trial was completed. The majority of the population (64%) are Muslim and 33% are Ethiopian Orthodox [26]. Trial baseline data revealed that approximately three quarters of the population in study villages owned latrines, but that most of these were low-quality unimproved pit latrines (Figure 1).

For this study, the definition of an improved latrine was: a pit-hole of 2 m depth or more; installation of a slab and a pit-hole cover; construction of a wall, door, and roof; and installation of a hand-washing facility with soap (Figure 2). A partially improved latrine was defined as a non-improved latrine that was nonetheless equipped with at least a pit, pit-hole cover, and slab. All the other latrines that did not meet the definition of an improved or partially improved latrine were defined as simple pit latrines.

### 2.2. Intervention

For the randomized trial, a “phase-in” design was adopted [27]. In the first phase, 24 randomly sampled villages within the two districts received a CLTS intervention between February 2016 and January 2017. The other 24 served as a control group in which nothing was done beyond health extension workers (HEWs) continuing their usual activities, which did not include substantial sanitation-related work (discussed in Appendix A). In the second phase from February 2017, the control villages received the same intervention. Enrolled households were followed up four times during 10 months.

CLTS was rolled out in the intervention group as follows. A group of CLTS facilitators was trained, comprising of district health officials and health professionals working for health centres, as well as HEWs from health posts. They undertook CLTS triggering in the 24 villages in February and March 2016. One or two CLTS promoters were recruited from every village, and they performed follow-up activities after the triggering by encouraging community members to build improved latrines through community conversations and household visits together with CLTS facilitators. No financial or material subsidies were provided for constructing household latrines in this intervention. After the intervention, 69% of households had a household latrine equipped with at least a pit, pit-hole cover, and slab, and 35% had an improved latrine; full data on latrine types and characteristics are reported in the main trial study, as well as more details of the setting and intervention [25]. The CLTS intervention was implemented as a sub-component of a larger project funded by KOICA, which included installation of gravity-fed piped water to public taps. Since water systems were installed after the CLTS intervention was completed, both in the intervention and control villages, the water component is of no relevance to the present analysis. We obtained ethical approval from the National Research Ethics Review Committee under the Ministry of Science and Technology, Federal Democratic Republic of Ethiopia (NRERC 3.10/032/2015; 29 July 2015) and the London School of Hygiene and Tropical Medicine (LSHTM Ethics Ref: 16260; 22 February 2019).

### 2.3. Study Design

Our study design comprised of an estimation of the incremental costs and benefits of a CLTS intervention compared to a “no intervention” scenario where HEWs continued their usual activities. Costs and benefits were modelled over a 10-year horizon, for the trial study population of 9713. We extrapolated data from a cluster-randomized controlled trial in which these scenarios comprise of the intervention and control groups, taking a societal perspective [10]. We followed reference case guidelines [28,29] and adhered to the Consolidated Health Economic Evaluation Reporting Standards (CHEERS) [30]—see Appendix A; Appendix A in the CHEERS statement consists of a 24 item checklist and provides guidance on the minimum amount of information required for reporting health economic evaluations (Appendix A). All analyses are reported in constant 2016 international dollars (Int’l $). Details of the equations underlying the CBA model are described below.

### 2.4. Benefit Measurements

The present value of total benefits (TB_PV_) is (Equation (1)):(1)$${\mathrm{TB}}_{\mathrm{PV}}=\sum_{t=1}^{T}\sum_{k=1}^{3}P_{\mathrm{tk}}\times B_{\mathrm{tk}}{\left(1+r\right)}^{-t}$$
where TB_PV_= present value of total benefits to all people in the intervention communities; B_tk_= value of the benefits to each member of age group k in year t; r = discount rate; P_tk_ = number of population in age group k and year t in the intervention communities; and T = lifespan of a household latrine.

The value of the benefits comprises of (a) benefits from premature deaths averted, (b) benefits from diarrhoea cases avoided, and (c) benefits from increased accessibility.

The value of the benefits (B_tk_) is (Equation (2)):B_tk_ = (PDA_tk_ × VSL_t_) + (DCA_tk_ × COI_tk_) + (TS_tk_ × VOT_tk_ × HI_tk_)(2)
where PDA_tk_ = premature deaths averted in age group k and in year t, corresponding to the number of deaths avoided due to the intervention’s effect; VSL_t_ = value of a statistical life in year t; DCA_tk_ = diarrhoea cases averted in age group k and in year t; COI_tk_ = the cost of illness in age group k and in year t; TS_tk_ = time savings in age group k and in year t, corresponding to the number of hours saved from no longer walking to an open defecation place (or a communal latrine, or a neighbour’s latrine) due to the intervention for each member of age group k in year t; VOT_tk_ = value of time for a member of age group k in year t, corresponding to a fraction of the average hourly income of the people in the intervention communities; and HI_tk_ = the average hourly income of the people in the intervention communities.

#### 2.4.1. Health Benefits

Health benefits were estimated as the value of (i) premature diarrheal deaths averted, (ii) cost of illness from diarrhoea cases averted. The ratio of longitudinal prevalence of child diarrhoea between intervention and control villages was estimated. Children under 5 in the study area were reported to have 5.4 days of diarrhoea in 365 days in the control group, using a novel diary method [24]. The longitudinal prevalence and duration of child diarrhoea were recorded by caregivers using the diary method for 140 days. Diarrheal calendars were distributed to 906 households in May 2016 and caregivers were requested to mark O or X on each date of the calendar according to the presence or absence of a daily diarrheal episode. The trial identified a 29% relative reduction in the longitudinal prevalence of diarrhoea in the intervention compared with the control villages, leading to 1.6 days of diarrhoea averted per child per year (Table 1).

Unlike many previous studies [13,14,15], we estimated the cases averted in other age groups based on primary data. The baseline survey reported that the ratios of 7-day diarrhoea prevalence among children aged 5–14 years (school-age children) and people older than 14 years (working age population) were 0.27 and 0.21, respectively, compared with children under 5 years of age. Based on these ratios, we estimated that the CLTS intervention led to a reduction of 0.42 and 0.33 days of diarrhoea per person per year among children aged 5-14 years and people older than 14 years, respectively.

(i) Benefits from premature deaths averted. To estimate deaths averted, we multiplied the total cases averted by the case fatality rate (0.07%, 0.05%, and 0.03% for each age group, respectively) [31]. To value avoided mortality, we used the value of a statistical life (VSL) method, as recommended by a CBA reference case [29,32]. This assumes that survival of a working-age individual would yield economic returns, and surviving children would yield economic returns after they reach a productive age. A benefit transfer approach was used to estimate the VSL for Ethiopians. A commonly used VSL for the USA is Int’l $8.9 million, where the gross national income (GNI) per capita was Int’l $58,700 in 2016, whereas the GNI per capita in Ethiopia was Int’l $1,730 [32,33]. With an income elasticity of 1.5, the VSL for Ethiopia in 2016 was estimated to be Int’l $45,194 [32].

The total premature deaths averted (PDA_t_**_k_**) is (Equation (3)):PDA_tk_ = DCA_tk_ × CFR_k_(3)
where DCA_tk_ = diarrhoea cases averted in age group k and in year t; and CFR_k_ = case fatality rate of diarrhoea for a person in age group k.

(ii) Benefits from diarrhoea cases avoided. The occurrence of fewer cases of diarrhoea can bring about economic benefits by reducing the costs of health and non-health care. The proportion of caregivers seeking treatment (i.e., taking their child to a health facility, drug store, or traditional healer) when their children contracted diarrhoea was assessed from trial data. The number of diarrhoea cases averted was multiplied by the unit cost of diarrhoea treatment from trial data. We assumed a zero benefit from reduced costs of self-treatment. Non-health-sector direct benefits relate to the reduced costs of transportation to health facilities and other resultant expenses such as meals and accommodation, which were assessed from trial data. Transportation costs to a drug store or a traditional healer were not estimated. When a child contracts diarrhoea, she or he needs to receive more care from caregivers, meaning they lose productive time [14,15,16]. Similarly, if an adult has diarrheal disease, she or he may lose the opportunity to engage in productive work [14,15,16]. This time has an opportunity cost. We valued an hour of time at 50% of the average hourly income of community members aged 15 or above, and 25% for school-age children between 5 and 14 years of age [16,34]. Household income of the study population was surveyed during the trial, and the mean was used to calculate the value of time, because the majority of people were not formally employed. The average hourly income of caregivers was multiplied by the total hours saved per year. We assumed 1920 working hours per year (240 days per year: 4 weeks per month; 5 days per week; and 8 h per day). The time spent by HEWs treating patients with diarrhoea was valued using the same method [16,20]; see more details in the Appendix A.

We estimated a 2% annual income growth for the base case, which was incorporated into the calculation of the VSL and the value of time [16]. Population growth was not incorporated, to allow comparison with the latest hypothetical CBA of a CLTS intervention [16]. We thus assume consistent size of population in each age group over the time horizon. However, as part of the sensitivity analysis, we ran a separate analysis taking population growth into consideration.

The benefit from diarrheal cases avoided (the cost of illness, COI_tk_) is (Equation (4)):COI_tk_ = P_tk_ × SMT_k_ × ((C_ipk_ × IP_k_) + (C_opk_ × OP_k_) + TP_k_ + M_k_ + (AC_k_ × IP_k_) + (HOP_k_ × VOT_tk_ × HI_tk_) + (HIP_k_ × VOT_tk_ × HI_tk_) + (HOP_k_ × VOTHEW_t_ × HIHEW_t_) + (HIP_k_ × VOTHEW_t_ × HIHEW_t_)) + P_tk_ × (1 − SMT_k_) × HNSMT_k_ × VOT_tk_ × HI_tk_(4)
where, SMT_k_ = percentage of diarrhoea cases for which individuals in age group k visit health facilities to seek medical treatment; C_ipk_ = cost of inpatient care; C_opk_ = cost of outpatient care; IP_k_ = percentage of diarrhoea patients visiting a health facility to seek medical treatment in age group k who receive inpatient care; OP_k_ = percentage of diarrhoea patients visiting a health facility to seek medical treatment in age group k who receive outpatient care; TP_k_ = transportation cost for those visiting a health facility to seek medical treatment; M_k_ = cost of food and drinks for those visiting a health facility to seek medical treatment; AC_k_ = accommodation cost for those visiting health facility to seek medical treatment; HOP_k_ = number of working hours lost due to being sick or caring for a sick person in age group k for those receiving outpatient care; HIP_k_ = number of working hours lost due to being sick or caring for a sick person in age group k for those receiving inpatient care; VOTHEW_t_ = value of time for HEWs in year t; HIHEW_t_ = the average hourly income of HEWs in year t; and HNSMT_k_ = number of working hours lost due to being sick in age group k for those not visiting health facilities.

#### 2.4.2. Time Savings from Increased Accessibility to a Household Latrine

We estimated how many households had switched from defecating in the open, a communal latrine, or a neighbour’s latrine to their own household latrine, and how much time they saved from the switch, based on trial data. These savings were monetized using the same value of time as above for all individuals aged over-5. The value of time savings was not counted for children under 5. As for the frequency of trips to an open defecation site, a communal latrine, or a neighbour’s latrine prior to the intervention, we assumed six times per day for women and once a day for men based on discussions with people in the community. Male members reported mostly urinating around the household compound, while women did not.

The benefit from increased accessibility (time savings, TS_tk_) is (Equation (5)):(5)$${\mathrm{TS}}_{\mathrm{tk}}=\sum_{m=1}^{M}{\mathrm{TS}}_{m}\times F_{k}\times 365$$
where, TS_tk:_ = time savings in age group k and in year t; TS_m_ = time saved per each community member M for age group k from not walking to an open defecation site, a communal latrine, or a neighbour’s latrine; M = number of community people who shifted from open defecation or communal latrine use to household latrine use in the intervention communities; and F_k_ = number of times a person defecates or urinates per day.

### 2.5. Cost Measurements

An incremental cost analysis [10] was used, in which the costs of the CLTS intervention were compared to the costs of the limited sanitation-related activities usually undertaken by HEWs (which characterizes what took place in control areas). The cost of the latter is estimated using a time and motion study of HEWs activities in Ethiopia [35,36,37,38,39]—full details are provided in Appendix A. The Ethiopian government submitted a proposal for the integrated water, sanitation, and hygiene (WASH) project to KOICA in 2013 and the target areas of the project were selected among the districts that had no previous CLTS intervention [40]. The SNNPR state has been implementing CLTS in eight districts since 2009, but the Cheha and Enemor Ena Ener districts had never previously received a CLTS intervention [41].

Since we took a societal perspective, we included the value of all resources for implementing and maintaining the CLTS intervention over the horizon, and other resultant costs. We followed reference case definitions of capital and recurrent costs [29]. Data were drawn from the project’s financial records and household survey results. Costs were categorized in four ways: initial investment, recurrent costs, program costs, and local investments. The costs of management, training for CLTS facilitators and CLTS promoters, community education, and incentives for CLTS promoters were categorized as program costs. Local actors’ and community members’ time spent on CLTS activities, including latrine construction, were categorized as local investments. If people purchased materials for latrine construction, those were also categorized as local investments. A top-down approach was used to estimate the costs of the program (based on financial records), while a bottom-up approach was used to estimate the costs of local investments (based on combining resource use estimates with unit costs).

The recurrent costs of latrines included maintenance, operations, and hygiene education, which were estimated as 10% of annualized capital costs in the base case [12,13,14,15,16,42]. Project financial records were audited by an independent accountant assigned by KOICA. The time horizon for estimating costs and benefits was 10 years, which is the estimated average useful life of an improved latrine in this setting, modified in sensitivity analyses (Appendix A). UNICEF estimated the lifespan of a pit latrine to be 10 years assuming a 2-m pit depth and six members in a household [43]. Hutton estimated the lifespan of a basic latrine to be 8 years and that of safely managed sanitation to be 20 years [44]. Considering the widely accepted parameter values for the lifespan of latrines by type, 10 years for an improved latrine in this study is an appropriate value, even when taking the possible abandonment rate into account (Appendix A) [44,45,46,47,48,49,50].

The present value of total costs (TC_PV_) is (Equation (6)):(6)$${\mathrm{TC}}_{\mathrm{PV}}=\mathrm{CRCP}+\mathrm{CRCCL}+\sum_{t=1}^{T}O\&M_{t}{\left(1+r\right)}^{-t}+\sum_{t=1}^{T}E_{t}{\left(1+r\right)}^{-t}$$
where, TC_PV_ = present value of total costs = initial cost + additional cost (operation and maintenance, and education); CRCP = capital cost and recurrent cost of program; CRCCL_:_ = capital cost and recurrent cost of community and local stakeholders’ commitment; O&M_t:_ = operation and maintenance cost in year t, 10% of annualized initial capital cost; E_t:_ = education cost in year t, 10% of annualized initial capital cost; capital cost items of program = cost for vehicles, motorcycles, and items worth more than US$100; capital cost items of community and local government commitment = cost for latrine construction; recurrent cost of program = cost for training, facilitation, management, and salary spent by the program management team; and recurrent cost of community and local stakeholders’ commitment = cost of time spent by local actors and community people (except for latrine construction).

The annualized investment cost (E) is (Equation (7)):E = (K − (S/(1 + r) n))/A(n,r)(7)
where, E = the annualized investment cost; K = the purchase price; S = the resale price (assumed to be 0); n = the lifespan of boreholes; r = the discount rate; and A(n,r) = the annuity factor, A(n,r) = (1 − (1 + r) − n)/r (n years at r discount rate).

The capital cost of latrine construction (CC) is (Equation (8)):(8)$$\mathrm{CC}=\sum_{h=1}^{H}{\mathrm{CCLM}}_{h}+\sum_{\mathrm{hh}=1}^{\mathrm{HH}}{\mathrm{TLC}}_{\mathrm{hh}}\times {\mathrm{VOT}}_{k=3}\times {\mathrm{HI}}_{03}$$
where, C = capital cost; CCLM_h_: capital cost for latrine construction materials purchased for a household h; H = number of households that purchased materials for construction of a household latrine; TLC_hh_ = time in hours spent on household latrine construction by a household hh; HH = number of households that constructed a household latrine; capital cost items of program (vehicles, motorcycles, and items worth more than US$100) = annualized cost × days of project implementation/365; and HI_03_ = hourly income for adults at time 0.

### 2.6. Sensitivity Analyses

To explore uncertainty surrounding estimates, we carried out one-way and probabilistic sensitivity analyses. The probabilistic sensitivity analyses explored parameter uncertainty. We conducted 10,000 Monte Carlo simulations, varying model parameters over a range of plausible values. The one-way analyses primarily explored methodological uncertainty around our assumptions about toilet useful life, the value of time, and the discount rate. We used 5 years and 15 years as the upper and lower bounds for the useful life of a latrine. For the value of time, a range of 25–75% was used for adults and 0–50% for school-aged children [34]. For the discount rate, 0% and 8% were used as the lower or upper bounds. Finally, in the base case, we assumed that intervention effects are sustained throughout the time horizon. In an alternative “slippage” scenario, we assumed that reversion to pre-intervention behaviours occurs at 3.5% a year. This assumption is half the 7% observed in a study of CLTS sustainability in Ethiopia by Crocker et al. because the latrine quality achieved in our study was higher and baseline proportion of open defecation practice was low in our setting [51]. We modelled this as a 3.5% annual decrease in cases averted, costs averted, and time savings. The parameter distributions for each variable in the probabilistic sensitivity analysis, the parameter range used for the one-way sensitivity analysis and its justification are described in the Appendix A. We present a tornado plot for the one-way analyses, and cumulative frequency distributions for the probabilistic analyses.

## 3. Results

The base-case values for the parameters of the benefits and costs of the CLTS intervention are presented in Appendix A. There were 1737 households in the 24 intervention villages. The number of people by age group was 1301 for under-5 children, 3804 for children aged 5–14, and 4608 for people aged 15 or above. Sixty-three percent of caregivers reported seeking health care when their child had diarrhoea, comprising of 56% who took their child to a health facility and the other 3% to a drugstore or a traditional healer. The other 4% sought home-based care. Among the children with diarrhoea, 5% were reported to have been hospitalized, for an average of 5 days. When the children had diarrhoea, they needed to receive 1 day of care from their caregivers. The proportion of open defecation declined by 3 percentage points (pp) among children aged 5-14 and 4 pp among people aged 15 or above, and people reported saving 9 min for each round trip from the switch. Twenty percentage points of people aged 15 or above reported switching from using a neighbour’s latrine to their own household latrine, which allowed them to save 5 min per round trip.

The number of diarrhoea cases avoided, premature deaths averted, and hours saved are presented in Table 2. We estimate that 20,374, 16,084, and 15,154 cases of diarrhoea would be avoided for each age group from under-5 children to the working-age population (aged 15 or above) over 10 years after the CLTS intervention in the intervention villages. Twenty-two premature deaths would be averted over the 10-year horizon. Furthermore, 412,893 h are projected to be saved from the avoided diarrhoea cases, and 2,064,902 h are expected to be saved by switching from open defecation or utilization of communal or neighbours’ latrines to a household latrine.

Table 3 summarizes benefits and costs by age group and overall. Avoided premature deaths accounted for 58% of the total benefits, followed by time savings from increased access to household latrines. The absolute value of health benefits was highest for under-5 children because almost 40% of the diarrhoea cases avoided and more than 60% of the premature deaths averted were estimated in this age group. Project implementation and management accounted for 54% of the total costs. The cost of community and local stakeholders’ investments in CLTS activities was Int’l $186,690, constituting 42% of the total costs. The benefit–cost ratio (BCR) was 3.7 and the net present value was Int’l $1,193,786. If we consider slippage, the BCR was 3.1 and the net present value Int’l $916,500 (Appendix A). The BCR was 4.3 and the net present value Int’l $1,453,794 when incorporating population growth (Appendix A). Appendix A visualize the distribution of benefits by item and age group, respectively. Appendix A shows that switching from open defecation, using communal latrines, and using neighbours’ latrines contributed to the benefits from increased accessibility to a similar extent. Time saved of caregivers and health professionals was the key component of benefits from avoided diarrhoea cases. For under-5 children, averted premature deaths were the key factor underlying the benefits, accounting for 83% (Appendix A). For the other age cohorts, increased accessibility was the main contributor to the benefits, accounting for 50% or above. In the base case, achieving these benefits assumes that the effects seen at trial endline (10 months after the CLTS triggering) are sustained throughout the 10 years.

Table 4 presents details of the initial costs of project implementation and management. Recurrent costs constitute the majority of total costs (94%) for CLTS implementation and project management. Monitoring and follow-up after the CLTS triggering accounted for 54% of the cost of the CLTS implementation. Salary and benefits for Korean and local management staff constitute 71% of the project management costs. Payment for Korean staff alone accounted for 54% of the project management costs.

Table 5 presents the initial costs borne by community members and local stakeholders in terms of time and material. Recurrent costs constitute 55% of the total costs. Of the recurrent costs borne by community members and local stakeholders, CLTS follow-up accounted for 59%, followed by CLTS triggering (24%).

Figure 3 and Figure 4 present the results of the one-way sensitivity analyses. The largest changes in benefit–cost analysis outcomes were yielded by the effects on diarrhoea of the CLTS intervention and the useful life of an improved latrine. The BCR with low effectiveness (lower limit of 95% the confidence interval) was 1.4, and the ratio with high effectiveness increased to 5.4. The net present value of the CLTS intervention ranged from Int’l $155,791 in the low-effectiveness scenario to Int’l $1,952,322 in the high-effectiveness scenario. Similarly, the BCR ranged from 1.9 to 5.4 and the net present value (NPV) from Int’l $394,821 to Int’l $1,954,045 as the lifespan of an improved latrine ranged from 5 and 15 years. VSL and the discount rate were the next most influential parameters. The changes VSL and the discount rate were the next most influential parameters. The changes in response to variation in other parameters in the BCR and NPV were minimal.

Figure 5 and Figure 6 present the results of the Monte Carlo simulations with the cumulative density functions of the BCRs and NPVs of 10,000 draws. The 5th and 95th percentile of NPVs were Int’l $406,017 and Int’l $1,977,960. The 5th percentile of the BCR was 1.9, while the 95th percentile was 5.4. The Monte Carlo analysis results showed that the NPV was not below zero under any plausible circumstances.

## 4. Discussion

This study suggests that a CLTS intervention could yield a favourable return on investment, with a base case BCR of 3.7 and NPV of Int’l $1.2 million over a 10-year time horizon. The probabilistic sensitivity analysis results of a Monte Carlo simulation indicated substantial uncertainty but BCRs consistently greater than one (95% CI 1.9–5.4).

Results for benefit–cost metrics in our study are similar to those of many existing hypothetical models and ex-post studies [12,13,20]. These are benefit–cost analyses of global level sanitation improvements or a pilot rural sanitation intervention in India, all of which were not a CLTS intervention [12,13,20]. In our study, we used empirical results of a randomized trial to provide many parameter values, including the effects of the CLTS intervention, baseline conditions, time savings, care-seeking behaviour, and the relative contribution of inpatient and outpatient care.

A recent cost–benefit study of a hypothetical CLTS intervention found that CLTS interventions were not as attractive as some previous studies suggested [16]. The authors argued that the high benefit–cost ratios or net benefits of the majority of existing models were mainly a result of not incorporating the costs of the time commitment of community members. We accounted for the investments of CLTS facilitators, CLTS promoters, and community members in terms of their time and material commitment, and the benefit–cost ratio and net benefits remain attractive in this setting.

It is worth noting that outcomes in our study are almost identical to those in hypothetical analysis for high-uptake villages with an externality by Radin and colleagues [16]. They defined high-uptake as a 35 pp coverage increase. In our trial, there was a 35 pp increase of improved latrine at 10 months after the CLTS triggering (Figure 1) [25]. In addition, partially improved latrine increased from 12% at baseline to 34% at 10 months. Thus, the coverage increase in our trial can be categorized as high-uptake as per Radin’s definition. Regarding the existence of an externality, we assessed the effect of a CLTS intervention on diarrhoea using an intention-to-treat analysis, regardless of community members’ ownership of a latrine [25]. This means that we estimated the benefits of the CLTS intervention based on the finding that a reduction in diarrhoea cases would occur community-wide in the villages that received the CLTS intervention. This is analogous to the externality in Radin’s study. Notably, the benefit–cost analysis was 3.7 in our analysis and 3.8 for the high-uptake villages with an externality in their analysis. The proportion of benefits from time saving is also similar between our trial and their estimation, 29% and 30%, respectively. All in all, the two studies show high consistency in the results.

The effect size of the CLTS intervention and the life of a latrine were established as the two most influential parameters in our estimation. In this study, a substantial share of benefits was attributable to the protective effects of the intervention against diarrhoea. Many recent trials, though not all, have suggested that rural sanitation improvements, some of which were CLTS, were not effective against diarrhoea [52,53,54,55,56]. However, we identified an effect similar to the results of recent systematic reviews on sanitation improvements [57,58]. Achieving a near universal coverage of improved latrines (pit latrines with a slab) based on the definition of Joint Monitoring Programme (JMP) of WHO/UNICEF [2], as opposed to the unimproved latrines typically achieved under CLTS, might have been the key factor underlying the reduction of the longitudinal prevalence of child diarrhoea in the intervention villages.

Notably, this study reported that the CLTS intervention yielded net economic benefits, even in the context of low prevalence of open defecation (OD) prior to the intervention. The proportions of households that had a simple pit latrine and safely disposed children’s faeces were already high. Previously, many studies based on hypothetical models assumed the majority of people to be defecating openly at baseline, leading to a high proportion of benefits originating from time savings. The substantial health benefit identified in the trial informing our study may have been due to the majority of service level transitions being from “unimproved” to “improved”, rather than from “OD” to “unimproved” as is often achieved with CLTS [6,59]. Furthermore, there was a significant reduction in the fly count around pit-holes in the intervention group compared with the control group [25]. Chavasse and colleagues found that diarrhoea was substantially reduced (period prevalence ratio of diarrhoea in the intervention group compared with the control group: 0.78, 95% CI 0.64–0.95, *p* = 0.01) after controlling flies [60]. If the CLTS intervention were implemented more intensively, leading to even higher coverage of improved latrines, we might expect greater benefits.

This study has some limitations. For VSL, we extrapolated the values using the VSL income elasticity. The limitation of this method is that changes in income elasticity could lead to a large difference in VSL values [29,32]. Thus, we included this parameter in our sensitivity analysis, and the results indicate that the CLTS interventions yield high returns on the investment even when assuming the lowest value of VSL.

When cross-checking parameter values, we found that the treatment and transportation costs reported by caregivers were higher than those suggested by government officials in the districts. Although it is plausible that caregivers had to make extra payments not related to official charges or fees, we used the values reported by government officials to make a conservative estimation of the benefits. When using the treatment and transportation costs reported by caregivers, outcomes slightly increased (Appendix A). We included all the expenses and salaries for the Korean staff in the cost estimation. The key task of the Korean staff was to manage the project and report monitoring results to the donor, the Korean government. We think that a CLTS intervention with similar intensity could be replicated without support from a foreign project manager. Correspondingly, the costs might be substantially reduced in future interventions. It was surprising that at baseline women were reported to travel to open defecation sites, a communal or a neighbour’s latrine six times per day to urinate and defecate. However, since time savings from increased accessibility are far from being the biggest driver of benefits in our study, it would not substantially affect our results if this was found to be overestimate. 

The context of the rural areas where we conducted the trial has typical features of remote areas of sub-Saharan African countries in terms of remoteness of villages, high prevalence of diarrhoea, and low socioeconomic status. However, sanitation coverage in the study area was already high at baseline, unlike the context of low baseline coverage in many previous trials. The coverage of a simple pit latrine was 73% even before the CLTS intervention started, but we found that many of the household latrines were poorly constructed and unhygienic. For this reason, the importance of improved latrines was emphasized in the intervention. Another characteristic of this study lies in the relatively small number of villages, which made it possible to conduct an intensive CLTS intervention in terms of supervision and management particularly during the follow-up period after the triggering. Thus, we anticipate that the results of this study might not be replicable in some rural areas. It might be difficult to encourage community members to construct improved latrines in CLTS interventions, particularly where open defecation practices are rampant. However, the increase in coverage of improved or partially improved household latrines was comparable to that reported in many previous trials.

We did not include other possible health and non-health benefits. There are a number of diseases that are thought to be transmitted via the faecal-oral, faecal-skin, and faecal-eye pathways, including helminths, schistosomiasis, and trachoma, to name a few [58,61,62,63,64]. A lack of access to sanitation indirectly contributes to undernutrition [58,65,66,67,68]. Improved sanitation is also reported to have an impact on factors related to well-being such as comfort, safety, dignity, privacy, convenience, and status although these are difficult to quantify and value [12,69,70,71]. These benefits are believed to have greater impact on women and vulnerable groups. Access to improved sanitation is also believed to contribute to school attendance and academic performance [72]. Poor management of human excreta causes environmental pollution [12]. All these outcomes would ideally be included in a CBA of a sanitation intervention. However, we could not include other benefits than diarrhoea in the absence of empirical data in our trial. Therefore, we infer that the true BCR and NPV would be higher than those we estimated in this study if measured comprehensively, and this possibility warrants future study.

## 5. Conclusions

The outcomes of benefit–cost analyses strongly depend on local conditions regarding key parameters. Our study shows that the benefits of a CLTS intervention in rural Ethiopia exceeded its costs by a reasonable margin in the base case and under plausible scenarios in sensitivity analyses. We recommend the importance of improved latrines should be highlighted in CLTS interventions, since substantial benefits come from their effect on diarrhoea reduction. If CLTS or sanitation interventions emphasizing improved latrines are implemented in areas of low latrine coverage and a high proportion of OD with similar intensity to this trial, the benefits would be more substantial than those we found in this study.

## Figures and Tables

**Figure 1 ijerph-17-05068-f001:**
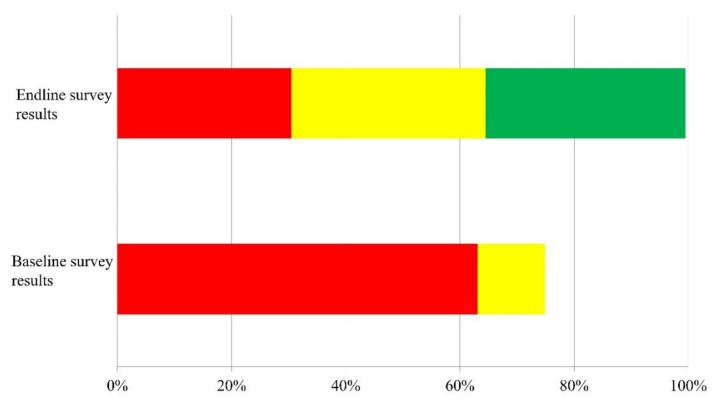
Sanitation coverage in the intervention group by the latrine type (red colour: simple pit latrine; yellow colour: partially improved latrine; and green colour: improved latrine. The white space in the bar denotes the proportion of households without any type of household latrine. The baseline survey was conducted in October–November 2015. Community-led total sanitation (CLTS) triggering was conducted in February to March 2016. The endline survey was carried out 10 months after the triggering).

**Figure 2 ijerph-17-05068-f002:**
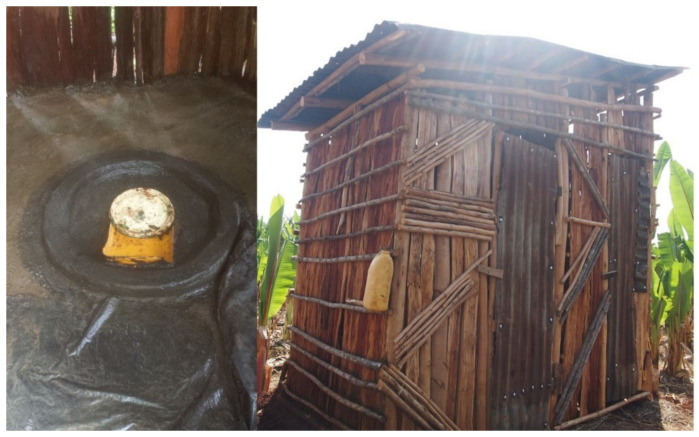
An improved latrine (left: inside, right: outside).

**Figure 3 ijerph-17-05068-f003:**
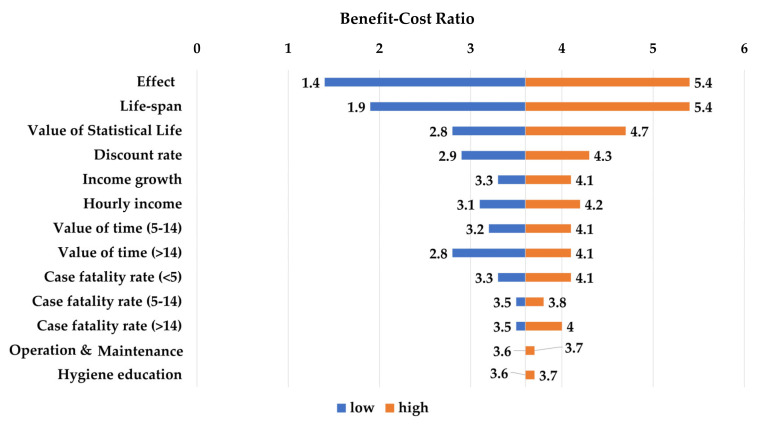
Results of the one-way analyses (benefit–cost ratio).

**Figure 4 ijerph-17-05068-f004:**
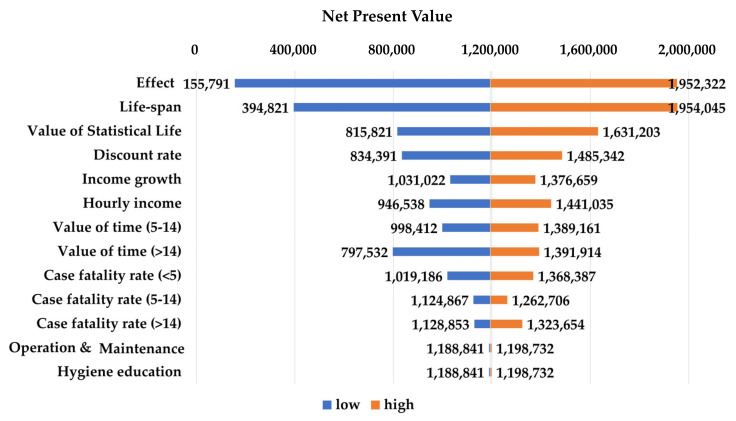
Results of the one-way analyses (net present value in 2016, Int’l $).

**Figure 5 ijerph-17-05068-f005:**
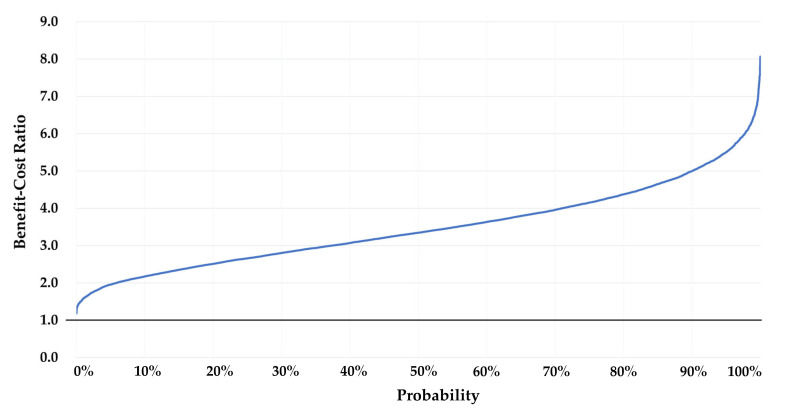
Cumulative probability of the benefit–cost ratio (Monte Carlo analysis, *x*-axis: cumulative percentage, and *y*-axis: benefit–cost ratio).

**Figure 6 ijerph-17-05068-f006:**
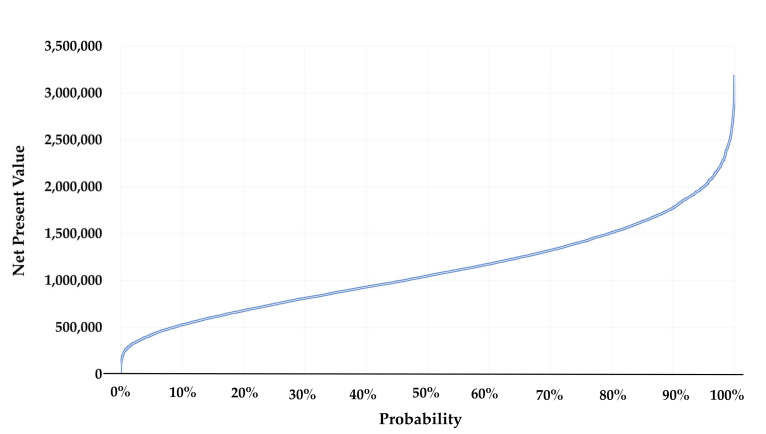
Cumulative probability of net present value (Monte Carlo analysis, *x*-axis: cumulative percentage, *y*-axis: net present value, and unit: Int’l $).

**Table 1 ijerph-17-05068-t001:** Effect of a CLTS intervention on the longitudinal prevalence of child diarrhoea.

	Intervention	Control
Total days of diarrhoea	481	773
Number of children	409	433
Person-days	49,571	52,467
Effect of CLTS on longitudinal prevalence of child diarrhoea (Reduced days of child diarrhoea per year) *	1.6 days, 95% Confidence Interval: 0.2–2.6 days *p* = 0.03	

* Adjusted for clustering effect and stratification.

**Table 2 ijerph-17-05068-t002:** Health and time benefits from the CLTS intervention.

	<5	5–14	≥15	SUM
Diarrhoea cases avoided	20,374	16,084	15,154	51,612
Premature deaths averted	14	3	5	22
Time saved from taking care of sick people (hours)	162,989	128,673	121,231	412,893
Time saved from the switch to a household latrine (hours)	0	1,101,556	963,346	2,064,902

**Table 3 ijerph-17-05068-t003:** Benefits and costs over 10 years (present value in 2016, Int’l $).

Item			Age Group				
			<5	5–14	≥15	Subtotal	%
**Benefits**	Avoided diarrhoea cases	Treatment costs saved	3294	1305	1230	5829	13%
		Transportation costs saved	3386	2312	2178	7876	
		Meal costs saved	903	465	439	1807	
		Accommodation costs saved	337	97	130	564	
		Caregiver time saved	51,769	20,435	38,506	110,709	
		Health professionals’ time saved	63,003	12,477	11,756	87,236	
		Subtotal	122,692	37,091	54,239	214,021	
	Averted premature deaths	Value of statistical life	611,103	137,839	194,801	943,744	58%
	Time saved from increased accessibility	Switch from open defecation	0	33,469	118,927	152,396	29%
		Switch from using communal latrines	0	46,949	91,930	138,878	
		Switch from using neighbours’ latrines	0	94,521	95,124	189,645	
		Subtotal	0	174,939	305,981	480,919	
	Total		733,795 (45%)	349,869 (21%)	555,021 (34%)	1,638,684 (100%)	100%
**Costs**	Initial costs						
	Project implementation and management	Recurrent				223,845	54%
		capital				14,580	
		Subtotal				238,425	
	Investment of community and local stakeholders	Recurrent				102,353	42%
		capital				84,337	
		Subtotal				186,690	
	Subtotal of initial costs					425,115	
	Operation and maintenance Education for the lifespan of a latrine	Operation and maintenance				9,892	4%
		Education				9,892	
		Subtotal				19,784	
	Total					444,899	100%
Benefit–Cost Ratio (BCR)			3.7				
Net Present Value (NPV)			1,193,786 (49,741 per community; 687 per household)				

**Table 4 ijerph-17-05068-t004:** Initial costs (program implementation and management, Int’l $).

		Item	Cost
CLTS ^a^ Implementation	Recurrent	CLTS promoter introduction	1200
		CLTS promoter training	2160
		Educating mothers	2160
		Community campaign	1800
		Information, Education and Communication Materials	9000
		Best promoter prize	2670
		CLTS training	5355
		CLTS implementation	2651
		Experience sharing	720
		Material incentives	3840
		Monitoring/follow-up after the CLTS triggering	43,204
		Meeting/workshop	4800
		Subtotal	79,560
Project Management	Capital	Motorcycle	5,590
		Vehicle	8,990
		Subtotal	14,580
	Recurrent	Korean staff salary and benefits	78,000
		Local management staff salary and benefits	24,840
		Translator	3600
		Stationery	4520
		Drivers	4800
		Fuel	12,000
		Office	3600
		Monitoring and evaluation	8925
		Report printing	4000
		Subtotal	144,285
Subtotal	Recurrent		223,845 (94%)
	Capital		14,580 (6%)
Total			238,425 (100%)

^a^ Community-Led Total Sanitation.

**Table 5 ijerph-17-05068-t005:** Initial costs (community members’ and local stakeholders’ investments, Int’l $).

	Item	Participants	Number of People	Hours/Person	Hourly Income ^a^	Cost
Recurrent ^c^	CLTS ^b^ training	District health officials	5	56	4.50	1260
		Health professionals (health centre)	5	56	2.54	711
		Health extension workers	24	56	1.79	2400
	CLTS promoter training	CLTS promoters	38	32	0.67	817
	CLTS orientation	District health officials	3	8	4.50	108
		CLTS promoters	38	8	0.67	204
	CLTS triggering	District health officials	5	40	4.50	900
		Health extension workers	24	192	2.54	11,704
		CLTS promoters	38	304	0.67	7740
		Community members	804	8	0.67	4311
	CLTS follow-up	District health officials	5	256	4.50	5760
		Health extension workers	24	256	1.79	10,998
		CLTS promoters	38	512	0.67	13,036
		CLTS committee	72	128	0.67	6175
		Community members	1079	32	0.67	23,127
		Kebele leaders	24	64	0.67	1029
	Review meeting	District health officials	12	64	4.50	3456
		Health extension workers	24	64	2.54	3901
		CLTS promoters	38	64	0.67	1629
		CLTS committee	72	64	0.67	3087
	Subtotal	-	-	-	-	102,353 (55%)
Capital ^d^	Latrine construction (time)	Community people	872	120	0.67	70,107
	Latrine construction (cement)	Community people	71	-	27.90	1968
	Latrine construction (handwashing facility)	Community people	721	-	17.01	12,263
	Subtotal	-	-	-	-	84,337 (45%)
Total		-	-	-	-	186,690 (100%)

^a^ in the case of latrine construction (for both cement and handwashing facility belong to capital item), it means unit price per household. ^b^ Community-Led Total Sanitation ^c^ Source of data: project report (monthly, annual, and final reports), household interview results. ^d^ Household interview results.

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
