# Peer review of "Benefits and Costs of a Community-Led Total Sanitation Intervention in Rural Ethiopia—A Trial-Based Ex Post Economic Evaluation"

_ijerph, 2020, doi:10.3390/ijerph17145068_

Round 1
Reviewer 1 Report
The article follows a well-used methodology and sufficient data coverage.
However, aspects of methodology selection could be more reasonable.
The design of the study presentation itself should be a clearer presentation. All graphs and tables are presented in one place. Some of them could be moved to the annex, but the most important ones need to be described in detail.
There are unclear places for text formatting. lines 100-118 in italic, why?
lines 120-128 enlarged font.
unclear bullets divider in section 2.4.1 (lines 141-152)
The final conclusions and future directions of the study should also be strengthened.
Reviewer 2 Report
The objective of the paper was to estimate the costs and benefits of a Community-Led Total Sanitation (CLTS) intervention using the empirical results from a cluster-randomized controlled trial in rural areas of Ethiopia. This paper responds to a need for more ex-post and cost benefit analysis (CBA) studies, and may be the first CBA of a CLTS intervention based on actual implementation.
I have one suggestion that I think would improve the impact of the paper. One case the authors make for this paper is how information from this paper could be utilized for future studies. I’m interested in more discussion in what parts of the study could be repeated or replicated elsewhere. Is it simply the equations for estimating costs and benefits in Table 1? Would other parameters need to be added in Table 2? Would any parameter values in Table 2 remain constant? I’d also like more discussion on if this context is a “normal” case for CLTS or an extreme case.
Here are my individual line comments:
Abstract
-L29: “favorable economic returns in the right conditions” – “right” is very ambiguous. Please provide more context on the conditions.
- Introduction
-L57: It is mentioned that other studies have emphasized that costs and benefits can be highly context-specific. Could you explain why results from an empirical study like this one would be beneficial for other cases, given that results can be highly context-specific?
- Materials and Methods
-L87: In Figure 1, in the remaining white space (e.g. between 75-100% in the baseline case), is that considered open defecation, or flush toilets to septic? Please explain.
-L141-170: Perhaps a formatting issue with the bullet points
-L218-222: First sentence introduces toilet useful life, value of time, and discount rate. Suggest the following sentences follow the same order (currently value of time, discount rate, toilet useful life).
-Table S2: Spell out any abbreviations that may not be common knowledge (e.g. CFR)
-Table 1: In value of benefits equation, suggest moving (c) to its own line for clarity
-Table 1: Suggest single line between section of (a) Total premature deaths averted and (b) benefits from diarrheal cases avoided. Same for between (b) and (c). As it currently stands, it’s a little hard to follow (e.g. COI equation from b appears to be in section of (a))
- Results
-Table 2: Why is there a separation between the last two sections of the table? They both seem to be for “other groups (5-14 or >15)”
-Figure 2: an issue with labelling (Figure 2b is labeled, but not Figure 2a)
-Table S3, Table S4, and Table S5: some numbers have commas, others do not
-Figure 5: The number of x-axis labels is a little distracting. Consider using labels every 5%.
Figure 6: Label incorrectly says “Figure 1”. Consider using labels every 5%.
- Discussion and 5. Conclusions
-One case the authors make for this paper is how information from this paper could be utilized for future studies. Similar to my comment in the introduction, I’m interested in more discussion in what parts of the study could be repeated or replicated. Is it simply the equations for estimating costs and benefits in Table 1? Would other parameters need to be added in Table 2? Would any parameter values in Table 2 remain constant?
-I’d also like more discussion on if this context is a “normal” case for CLTS or an extreme case?
-L387: “effectively” and “right” are ambiguous—please define with more clarity
Reviewer 3 Report
The manuscript presents the benefits of using latrines in rural areas in Ethiopia. The work is of interest in the context of health improvement in study sites; but it needs to improve its overall presentation and better organize the presentation of the work.
The aspects that need to be improved are indicated in my opinion:
Figure 1. The caption should improve the wording, so that the description of the types of latrines should be in the text of the paper (section 2.2), not in the caption of the figure. However, it should indicate the meaning of Endline and Baseline.
Line 109-110. Figures S1 and S2 can be included in the text, one side by side, it is not necessary to place it as supplementary material.
Font regularization, no italics or large letters necessary.
Line 134-135, explain the method used briefly.
Table S1 should be included in the main text; its reduced length makes it suitable for placement in the results section.
Table 2, given its length, should be presented in the accompanying material.
Do not confuse the annexed material, which is placed at the end of the main text, with the supplementary material, which is published in a separate file from the main text file. In this sense, short results tables, such as the current S1, should be included in the main body with the results; the current Table 2, given its extension, should be included as an annex; the tables S3 and following, should be clearly shown as supplementary material.
The format in which the calculation equations are presented is not appropriate, each equation should have its equation number and the meaning of the abbreviations used in it should be briefly described below. It should be presented not as a Table, but as text in which each equation is a paragraph heading.
Lines 259-281. The figures should be interspersed with the explanatory texts of the results, rather than placing the whole text together before and the figures after. The description of the results should explain in the text the most interesting and significant details of the work; in some texts it is limited to pointing out the figure or the table, such as lines 271-273. It is advisable to highlight in the text what is most interesting about these results.
As for the bibliography, it is adequate; but the format used does not follow the rules of the journal. Most are incorrectly formatted. It is important to verify the correct handwriting of the authors, the use of journal titles in italics and abbreviated, the year in bold, the use of commas and periods.
As for the supplementary texts S1, S2, S3, their meaning in the context of the work should be highlighted in the results section (or methodology if applicable). While Text S1 is presented, texts S2 and S3 are purely a quotation in one point, without indicating what importance they have in the work.
Round 2
Reviewer 3 Report
The authors have made the proposed corrections and the manuscript is acceptable. I have only noticed one detail that can be modified: In table 4 and 5 it would be more correct to say instead of "Sum" and "Grand sum", put Subtotal and Total.